# PTRNet: Global Feature and Local Feature Encoding for Point Cloud Registration

Cuixia Li [1,2], Shanshan Yang [2], Li Shi [3], Yue Liu [2] and Yinghao Li [2,*]

1   School of Electrical Engineering, Zhengzhou University, Zhengzhou 450001, China; lcxxcl@zzu.edu.cn
2   School of Cyber Science and Engineering, Zhengzhou University, Zhengzhou 450001, China; youth_wassup@163.com (S.Y.); ly16637857359@163.com (Y.L.)
3   Department of Automation, Tsinghua University, Beijing 100084, China; shilits@tsinghua.edu.cn
*   Correspondence: yinghaoli@zzu.edu.cn

**Abstract:** Existing end-to-end cloud registration methods are often inefficient and susceptible to noise. We propose an end-to-end point cloud registration network model, Point Transformer for Registration Network (PTRNet), that considers local and global features to improve this behavior. Our model uses point clouds as inputs and applies a Transformer method to extract their global features. Using a K-Nearest Neighbor (K-NN) topology, our method then encodes the local features of a point cloud and integrates them with the global features to obtain the point cloud's strong global features. Comparative experiments using the ModelNet40 data set show that our method offers better results than other methods, with a mean square error (MSE), root mean square error (RMSE), and mean absolute error (MAE) between the ground truth and predicted values lower than those of competing methods. In the case of multi-object class without noise, the rotation average absolute error of PTRNet is reduced to 1.601 degrees and the translation average absolute error is reduced to 0.005 units. Compared to other recent end-to-end registration methods and traditional point cloud registration methods, the PTRNet method has less error, higher registration accuracy, and better robustness.

**Keywords:** point cloud registration; local feature; global feature; Transformer; K-Nearest Neighbor

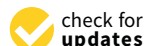



## 1. Introduction

In recent years, with the increasing maturity of laser technology, a variety of laser systems play an important role in cultural relic restoration, military aerospace, 3D topography measurement, target recognition and other applications. As one of the key technologies of laser 3D scanning and imaging, 3D point cloud registration has been widely studied and applied. Rigid point cloud registration refers to the problem of finding a transformation matrix T to align two point clouds. Registration plays a key role in many computer vision applications, such as 3D reconstruction, 3D positioning, and attitude estimation.

Traditional point cloud registration methods are often called optimization-based point cloud registration frameworks. Such registration algorithms [1–7] obtain the optimal transformation matrix by iterating through two phases [8] correspondence search and transformation estimation. The correspondence search finds the corresponding (matching) points between the point clouds. Transformation estimation uses these corresponding relations to estimate the transformation matrix. The advantage of these methods is that they do not need training sets, relying on strict data theory to ensure their convergence. However, they need many complex strategies to overcome noise, outliers, density changes, and partial overlaps, all of which increase the computational cost.

In recent years, the development of deep learning technology has enabled 3D point cloud registration to make great progress [9–14]. Point cloud registration methods based on deep learning fall into two categories: feature learning methods [15–19] and end-to-end methods [20–26]. Feature learning methods use depth features to estimate accurate

correspondences. After finding matching points, the transformation is estimated using one-step optimization (e.g., singular value decomposition (SVD) or random sample consensus (RANSAC)) without iteration between correspondence estimation and transformation estimation. Wang et al. [16] estimated the correspondence of depth features based on the traditional iterative registration method of iterative closest point (ICP) [1] and used SVD to calculate the transformation to complete registration. Yan et al. [17] used a differentiable Sinkhorn layer and annealing algorithm to determine corresponding points by learning the fusion features between spatial features and local geometric information. Li et al. [20] proposed an iterative distance-aware similarity matrix convolution network, which could be easily integrated with traditional feature methods (e.g., Fast Point Feature Histograms, FPFH [27]) or learning-based methods to perform registration. This approach provides robust and accurate correspondence searches, obtaining accurate registration results through a simple RANSAC (SVD) method. However, the registration performance of such methods declines sharply in unforeseen situations. If there is a large difference in the distributions between the actual scene and the training data, a large amount of training data is needed. Meanwhile, the separated training process learns a stand-alone feature extraction network, with the learned feature network only determining matching point pairs without the ability to complete the registration.

The end-to-end learning-based methods solve the registration problem with a single neural network. Such methods accept two point clouds as inputs and produce the transformation matrix between the two point clouds as output. These methods transform the registration problem into a regression problem. Deng et al. [11] proposed a local descriptor suitable for this task along with a network RelativeNet for relative pose regression on the basis of point cloud registration and point cloud relative pose estimation. Sarode et al. [21] proposed an end-to-end lightweight universal point cloud registration model with good performance in inheritability. Pais et al. [22] proposed an end-to-end deep learning architecture for 3D point cloud registration and compared the results of SVD and neural networks based on regression problems, with experiments showing the latter to be faster and more accurate. These end-to-end registration methods have the advantage of being designed and optimized specifically for point cloud registration using neural networks, while retaining the advantages of both traditional mathematical theory and deep neural networks. However, the distance measurement between point clouds is measured in Euclidean space, with its transformation parameter estimation regarded as a "black box" with no traceability. In addition, these methods often ignore local information in the point clouds. Although end-to-end point cloud registration using deep learning has become a primary research topic, effectively extracting point cloud features conducive to registration and carrying out robust, efficient, and accurate registration remain issues to be studied.

To overcome these problems, we propose an end-to-end registration model called Point Transformer for Registration Network (PTRNet) that takes both local and global features into account. Using a KNN topology, this method encodes point clouds locally and uses an attention mechanism to distinguish the importance of the point cloud features. Doing so extracts more powerful global information. Our experimental results show that PTRNet offers good robustness and high registration accuracy.

## 2. Method

As shown in Figure 1, the PTRNet model consists of an encoder and a decoder. Point cloud data are first encoded and then inputted into an encoder, which converts the input points into high-dimensional spatial features. Mapping the points into a high-dimensional space simplifies the extraction of the local and global features of the point cloud. The second part of the task uses the two features from the encoder as input for point cloud registration. Specifically, the six-layer fully connected network predicts the relative pose, combining the translation and rotation quantities and outputs seven data elements: three translation quantities and four unit quaternions obtained through normalization.

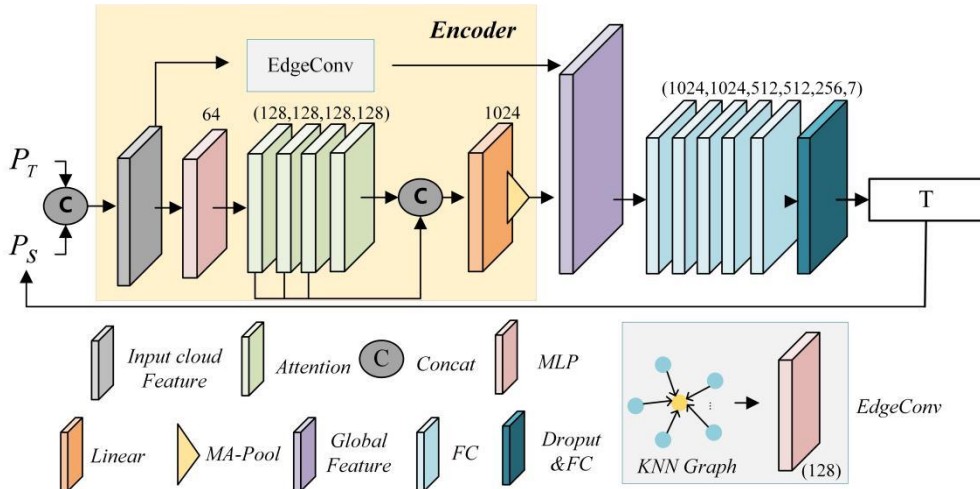

**Figure 1.** The structure of PTRNet.

*2.1. Local Features*

To better capture the local geometric features of the point cloud while maintaining the invariance of the arrangement, PTRNet first constructs adjacent geometric graphs and then convolves the edges of the local graphs, collecting local features of the point cloud for subsequent registration [28]. The specific operations are as follows.

We consider a point cloud with $N$ points and $F$ dimensions at each point, denoting the set. $X = \{x_1, x_2, \ldots, x_N\} \subseteq \mathbb{R}^F$. When $F = 3$, each point contains three dimensional coordinates $x_i = (x_i, y_i, z_i)$. When $F > 3$, the information for each point contains additional data indicating colors, surface normals, and so on.

We establish the KNN topology as shown in Figure 2. The graph $G = \{V, E\}$ represents a point $x_i$ and its $k$ neighboring points $\{x_{j1}, x_{j2}, \ldots, x_{jk}\}$, where $V = X$, $E = < i, jk >$, and $x_i, x_{jk} \in V$. As a directed graph of the local point cloud structure, we define the edge feature as $e_{ij} = h_{\Theta}(x_i, x_j)$, where $h(\cdot) : \mathbb{R}^F \times \mathbb{R}^F \to \mathbb{R}^F$ is a nonlinear function with learning parameter $\Theta$ and $F'$ is the number of dimensions of the edge features. The $K$ edge features are aggregated into $\varsigma$ (i.e., accumulation or maximization) to obtain the output $y_i$ corresponding to $x_i$, which is defined as Formula (1).

$$y_i = \underset{j:(i,j) \in E}{\varsigma} \, h_{\Theta}(x_i, x_j) \tag{1}$$

During the convolution operation in the images, $x_i$ denotes the central pixel and $x_j$ the pixel block around $x_i$. If the input of this layer is an F-dimensional point cloud of N points, it is output as $F'$-dimensional edge features of N points.

The choices of edge function and aggregation method are key to the KNN graph convolution network [28]. The edge function should consider the local characteristics of the point cloud, so we define the edge function as

$$h_{\Theta}(x_i, x_j) = h_{\Theta}(x_i, x_i - x_j) \tag{2}$$

In (1), a feature is considered as a set of multiple patches. The global information of (2) is obtained by a function based on query point $x_i$, with the local information is obtained by $(x_i - x_j)$ to obtain stronger feature representations. Using new symbols, we define the resulting feature equation as

$$e'_{ij} = ReLU(\mu \cdot x_i + \lambda(x_i - x_j)) \tag{3}$$

The implementation could be seen as a multilayer perceptron (MLP), with the learning parameter defined as $\Theta = \{\mu, \lambda\}$ and the activation function defined as *ReLU*. In graph

convolutions, max or mean functions are often chosen arbitrarily as aggregation functions. We adopt a max function for aggregation operations. Edge features are aggregated by (4):

$$y'_i = \max_{j:(i,j)\in E} e'_{ij} \tag{4}$$

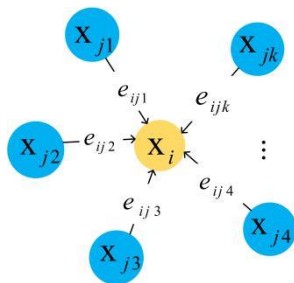

**Figure 2.** KNN graph convolution operation.

### 2.2. Transformer

Attention mechanisms use relative importance to focus on different parts of the input sequence, highlighting the relationship between inputs enabling the capture of context and high-order dependencies. Inspired by the point cloud processing Transformer proposed in recent years [29,30], we define Q, K, and V as the query, key, and value matrices, respectively, generated by the linear transformation of input characteristics. $F_{in} \in \mathbb{R}^d$ The function $A(\cdot)$ describes the mapping of N queries $Q \in \mathbb{R}^{N \times d_k}$ and N key values to $K \in \mathbb{R}^{N_k \times d_k}$ and $V \in \mathbb{R}^{N_k \times dv}$ to the output [31]. The attention weight is calculated by the matrix dot product $QK^T \in \mathbb{R}^{N \times d}$:

$$score(Q, K) = \sigma(QK^T) \tag{5}$$

where $score(\cdot): \mathbb{R}^{N \times d_k}, \mathbb{R}^{N_k \times d_k} \to \mathbb{R}^{N \times N_k}$, and $\sigma(\cdot)$ is the activation function $softmax(\cdot)$. The attention function is obtained by weighting the value of (5) according to V:

$$A(Q, K, V) = score(Q, K)V \tag{6}$$

where the output $A(\cdot): \mathbb{R}^{N \times d_k}, \mathbb{R}^{N_k \times d_k}, \mathbb{R}^{N_k \times dv} \to \mathbb{R}^{N \times d_k}$ is equivalent to a weighted sum of V. If the dot product between the key and the value yields a higher score, the weight of the value is greater. The model dimensions are set to $d_k = d_q = d_v$ if not specified.

Our graph convolution network research studied the benefits of using a Laplace matrix [32] $L = D - E$ instead of an adjacency matrix $E$, where $D$ is the antiangular matrix. Using the former, the output characteristics of the offset-attention used by PCTNet [29] are

$$F_{out} = LBR(F_{in} - F_a) + F_{in} \tag{7}$$

whereas *LBR* represents three operations: linear operation, batch processing, and the *ReLU* activation function. $F_a = A(Q, K, V)$ and $F_{in} - F_a$ are similar to a discrete Laplace operator [29]. Our experiments showed that self-attention was more beneficial for subsequent registration tasks when replaced by offset-attention.

### 2.3. PTRNet for Point Cloud Registration

As shown in Figure 1, the encoder partially extracts local and global features. The concatenation operation integrates the two features into a global feature vector $F_{global}$ Since $F_{global}$ contains geometric and direction information of the source and template point clouds, the transformation information can be extracted from the global feature vector through a series of operations expressed as

$$F_{P_S}, F_{P_T} = \varphi(F_{global}) \tag{8}$$

where $\varphi(\cdot)$ denotes a symmetric function, $P_S$ and $P_T$ represent the source and template point clouds, respectively, and $F_{P_S}$ and $F_{P_T}$ describe the global characteristics of the source and template point clouds. Next, we calculate the rigid body transformation $T \in SE(3)$ to minimize the difference between $F_{P_S}$ and $F_{P_T}$. As shown in Figure 1, the global features are concatenated as input to the fully connected layer. The specific implementation adopts six full connection layers, with 1024, 1024, 512, 512, 256, and 7 channels. The seven parameters of the final output represent the estimated transform $t$. The transformation $T$ will be transferred in one direction in the network as the transformation estimation adjusts to the source point cloud $P_S$. The transformation $T$ obtained in each iteration adjusts the pose of the source point cloud $P_S$, and the adjusted source and template point clouds $P_T$ are sent to the network for the next iteration.

*2.4. Loss Function*

The loss function used to train the registration network should minimize the distance between the corresponding points in the source and template point clouds. This distance is calculated using a Chamfer distance metric:

$$
\begin{aligned}
CD(P_S^{est}, P_T) = {} & \frac{1}{P_S^{est}} \sum_{x \in P_S^{est}} \min_{y \in P_T} \|x - y\|_2^2 \\
& + \frac{1}{P_T} \sum_{y \in P_T} \min_{x \in P_S^{est}} \|y - x\|_2^2
\end{aligned}
\tag{9}
$$

where $P_T$ denotes the template point cloud, and $P_S^{est}$ denotes the source point cloud $P_S$ after the $T$ transformation. Finding an optimal location minimizes the distance between corresponding points.

**3. Experiments**

To verify the effectiveness of our PTRNet model, we conducted comparative experiments on the ModelNet40 dataset [33], which comes from the ModelNet dataset [33] and contains 12,311 meshed CAD models in 40 categories. It is a large dataset released in recent years and is widely used in point cloud processing tasks [16,21,25,34]. Following the experimental setup of PCRNet [21], we randomly selected 5070 transformations with Euler angles in the range [−45, 45] and translation values in the range [−1, 1] units. We sampled 1024 points uniformly from the outer surface of each model, with no other features in the input other than (*x*, *y*, *z*) coordinates.

We used the mean square error (MSE), root mean square error (RMSE), and mean absolute error (MAE) between the true and predicted values as metrics. Ideally, all of these error indicators should be zero if the source and template point clouds are perfectly registered. All angle measurements in the results are in degrees. We conducted our experiments on a computer using an Intel Xeon E5-2600 processor running Linux 18.4. We trained all networks to 300 epochs using a batch size of 1 for DCP_V2_MLP and 10 for all others.

We compared the performance of PTRNet with other recent end-to-end registration methods based on deep learning, iterative PCRNet [21] and DCP_V2_MLP [15], as well as the manual matching algorithm ICP [1]. In addition, we performed ablation experiments called Model-V1 and Model-V2 using Model-V1 to replace offset-attention with self-attention in PTRNet while leaving other parts unchanged. Model-V2 does not add local features to global features; it makes no other changes.

**4. Results**
*4.1. Training and Testing on Different Object Classes*

To test the universality of different models, we trained DCP_V2_MLP, iterative PCR-Net, ICP, Model-V1, Model-V2, and PTRNet on the first 20 categories of ModelNet40, for a total of 5070 models. We then selected 100 models for testing from 5 object categories that

were not in the training categories and without noise in the point clouds. We used the same source and template point clouds when testing all algorithms for a fair comparison.

The results for point cloud registration in the noise-free categories are shown in Table 1. PTRNet had the better registration results than either the depth or of traditional methods. It is worth noting that the iPCRNET model does not contain the Transformer module and the local feature extraction method adopted by us. As mentioned in Section 3, the ablation experiment Model-V2 adopts the same Transformer module as PTRNET but does not contain local features. Model-V1 contains local features but uses transformer modules that replace offset-Attention with self-Attention. For all performance indicators, the point cloud registration effect of models Model-V2 and Model-V1 with only local features or traditional Transformer is better than that of other algorithms, and this indicates that taking local features and fully considering global features in the stage of point cloud feature extraction will be more conducive to the later point cloud registration task. At the same time, PTRNet showed stronger performance with lowest mean error. On the other hand, DCP-V2 lost its registration ability when replacing SVD with MLP, which we attribute to the poor integration of the feature extraction network before and MLP after, while PTRNet was better as an end-to-end network. Figure 3 shows the point cloud registration for some representative objects, and it can be seen from the figure that point clouds after iPCRNet registration are scattered and that PTRNet has a better registration effect.

**Table 1.** Multiple Object Classes Without Noise.

| Model | MSE (R) | RMSE (R) | MAE (R) | MSE (t) | RMSE (t) | MAE(t) |
|---|---|---|---|---|---|---|
| iPCRNet | 9.521 | 3.086 | 3.811 | 0.793 | 0.891 | 0.025 |
| DCP-V2-MLP | 682.279 | 26.120 | 22.603 | 0.015 | 0.122 | 0.094 |
| ICP | 12.314 | 3.509 | 5.387 | 9.604 | 3.100 | 0.063 |
| Model-V1 | 4.874 | 2.208 | 2.083 | 0.022 | 0.148 | 0.011 |
| Model-V2 | 7.262 | 2.695 | 3.296 | 0.034 | 0.184 | 0.012 |
| PTRNet | 4.434 | 2.106 | 1.778 | 0.003 | 0.055 | 0.005 |

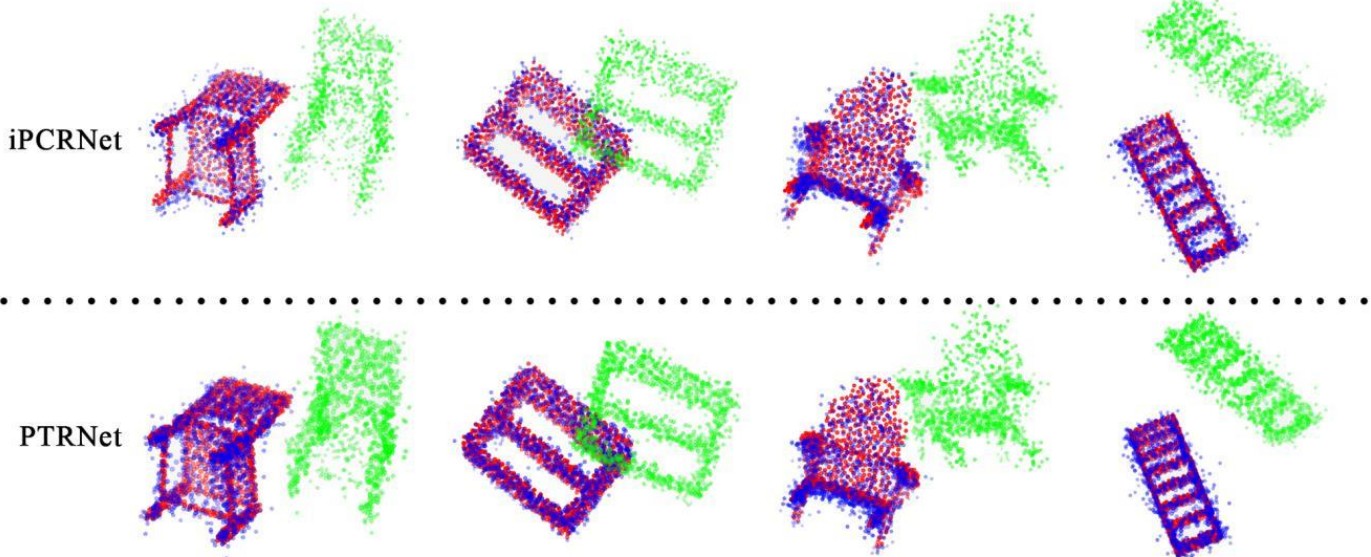

**Figure 3.** Point cloud registration results for iPCRNet and PTRNet: Red represents the template point cloud, green represents the source point cloud, and purple represents the registration result.

### 4.2. Training and Testing on the Same Object Class

In this experiment, we repeated the experiment in 4.1 and trained the network with objects from the same category as the test data. The PTRNet registration results greatly improved, with the average rotation error MAE (R) increasing from 1.778 to 1.601, the

average translation error MAE (T) remaining at a minimum of 0.005, and registration results maintaining relatively good results. At the same time, we noticed that the average errors (MAEs) of Model-V2 decreased significantly compared with experiment 4.1, and the MAEs of Model-V2 did not decrease significantly, which indicates that local information has less influence on registration results than global information in processing registration tasks of the same category. However, no matter whether it is Model-V1 or Model-V2, their MAEs is still lower than that of iPCRNet, which does not take sufficient account of local and global features. In addition, it also reflects the superior point cloud feature extraction capability of the offset-attention-based Transformer module we adopt. The experimental results are shown in Table 2.

**Table 2.** Same Object Category Without Noise.

| Model | MSE (R) | RMSE (R) | MAE (R) | MSE (t) | RMSE (t) | MAE(t) |
|---|---|---|---|---|---|---|
| iPCRNet | 1.220 | 1.104 | 2.715 | 7.844 | 2.801 | 0.006 |
| DCP-V2-MLP | 919.594 | 30.325 | 24.013 | 0.046 | 0.214 | 0.170 |
| ICP | 9.638 | 3.105 | 4.587 | 6.509 | 2.551 | 0.017 |
| Model-V1 | 4.180 | 2.045 | 2.057 | 0.023 | 0.152 | 0.007 |
| Model-V2 | 1.315 | 1.146 | 1.976 | 0.158 | 0.397 | 0.010 |
| PTRNet | 1.143 | 1.069 | 1.601 | 0.003 | 0.055 | 0.005 |

### 4.3. Robustness Experiment

To evaluate the robustness of the network to noise, we performed experiments on the point clouds with Gaussian noise. Using the dataset described in Section 4.1, we sampled the Gaussian noise independently from N (0, 0.01), clipped the noise to [−0.05, 0.05], and added it to source point clouds during testing. In this experiment, we trained the iterative PCRNet, DCP-V2-MLP, Model-V1, and object categories with noise.

We compared the accuracy of ICP, iterative PCRNet, DCP-V2-MLP, Model-V1, and Model-V2 with noise in the source point clouds to ensure that the data set had the same source and template point cloud pairs for the sake of a fair comparison. As shown in Table 3, the registration errors of each method are affected to varying degrees. Compared with experiment 4.1, the MAEs of iPCRNet and ICP methods are reduced, among which iPCRNet, which may benefit from the lightweight characteristics of its network, while ICP has reduced its error in this experiment, but it is the same as the DCP-V2-MLP, their registration effect is not ideal, which may be due to the fact that the ICP method easily falls into local optimum.. At the same time, the registration effects of Model-V1, Model-V2 and PTRNet are all disturbed by noise, which we think may be related to the complexity of the network. However, the MAEs of our model were the smallest.

**Table 3.** Multi-Object Classes With Noise.

| Model | MSE (R) | RMSE (R) | MAE (R) | MSE (t) | RMSE (t) | MAE(t) |
|---|---|---|---|---|---|---|
| iPCRNet | 1.973 | 1.405 | 3.518 | 1.863 | 1.365 | 0.013 |
| DCP-V2-MLP | 779.542 | 27.920 | 23.173 | 0.084 | 0.290 | 0.225 |
| ICP | 9.638 | 3.105 | 4.587 | 6.509 | 2.551 | 0.017 |
| Model-V1 | 7.003 | 2.646 | 2.907 | 0.239 | 0.489 | 0.015 |
| Model-V2 | 4.273 | 2.067 | 4.012 | 0.000 | 0.010 | 0.020 |
| PTRNet | 1.340 | 1.158 | 2.739 | 0.036 | 0.190 | 0.011 |

Table 4 shows the training test results on the same object category. In this test, the data set described in Section 3 was used, with Gaussian noise added to the source point cloud as described above. We trained the networks on a specific object class as in other research [21], and tested them on the same category using 150 vehicle types. Gaussian noise was added to the source point cloud during training and testing. As shown in Table 4, compared with multi-object classes with noise, the errors of each method are reduced. It is worth noting that the rotation error and translation error of Model V2 are reduced by 47%

and 40%, respectively, far more than that of Model-V1, which also confirms the viewpoint in Experiment 4.2. That is, local features have less effect on the same category registration than global features, and PTRNet had the strongest resistance to Gaussian noise and the best registration results.

**Table 4.** Same Object Category With Noise.

| Model | MSE (R) | RMSE (R) | MAE (R) | MSE (t) | RMSE (t) | MAE(t) |
|-------|---------|----------|---------|---------|----------|--------|
| iPCRNet | 1.274 | 1.129 | 2.771 | 5.700 | 2.388 | 0.016 |
| DCP-V2-MLP | 679.070 | 26.059 | 22.583 | 0.139 | 0.373 | 0.295 |
| ICP | 10.446 | 3.232 | 4.760 | 0.000 | 0.010 | 0.015 |
| Model-V1 | 1.134 | 1.065 | 2.767 | 0.193 | 0.439 | 0.006 |
| Model-V2 | 3.315 | 3.027 | 2.125 | 0.018 | 0.134 | 0.012 |
| PTRNet | 1.242 | 1.821 | 1.936 | 0.001 | 0.032 | 0.006 |

### *4.4. Efficiency*

We describe the test times of different methods on a server with an Intel E5-2600 CPU, an Nvidia GeForce RTX 2080Ti GPU, and 64GB of memory. The calculation time is calculated in milliseconds, with an average of 100 point cloud pairs containing 1024 points. As Figure 4 shows, iPCRNet is the fastest method, with PTRNet second only to iPCRNet. The reason is that the model complexity of iPCRNet, PTRNet and DCP-V2-MLP is from low to high, respectively. Meanwhile, the ICP algorithm requires every point on the data point cloud to find the corresponding point by traversing every point on the model point cloud, so its registration speed is slow.

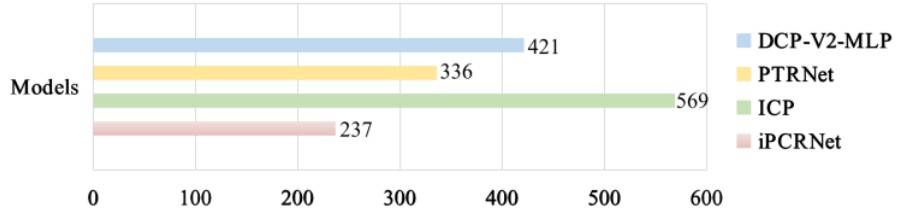

**Figure 4.** Test time (ms) of different methods.

### 5. Conclusions

In this paper, we have presented a new end-to-end point cloud registration network that considers both local and global characteristics of point clouds. By combining local feature encoding and an attention module, our model effectively extracts the corresponding information of the input point cloud, with the learned features greatly enhancing the point cloud registration. In addition, our method shows strong performance on multiple tasks. Especially in the case of the multi-object class without noise, the rotation average absolute error of PTRNet is reduced to 1.601 degrees and the translation average absolute error is reduced to 0.005 units. Experimental results prove that the proposed method is effective and robust compared with the contrast method. Furthermore, PTRNet is easily integrated with other processes and adaptable to many needs. However, the PTRNet model has high complexity, and the registration speed needs to be optimized. In the future, we plan to modify the network and simplify the model, so as to complete point cloud processing tasks such as cross point source registration and collective registration of multiple point clouds on the basis of PTRNet while realizing efficient registration.

**Author Contributions:** Writing—review and editing, C.L. and L.S.; writing—original draft preparation, S.Y.; visualization, Y.L. (Yue Liu); supervision Y.L. (Yinghao Li). All authors have read and agreed to the published version of the manuscript.

**Funding:** This research was funded by [the Network Collaborative Manufacturing Integration Technology and Digital Suite Research and Development Project of the Ministry of Science and Technology] grant number [2020YFB1712401]; [the Key Scientific Research Project of Colleges and Universities in Henan Province] grant number [21A520042]; [Major public welfare projects in Henan Province] grant number [201300210500] And The APC was funded by [the Network Collaborative Manufacturing Integration Technology and Digital Suite Research and Development Project of the Ministry of Science and Technology].

**Institutional Review Board Statement:** Not applicable.

**Informed Consent Statement:** Not applicable.

**Data Availability Statement:** Datasets that support reporting results can be found in the link: https://drive.google.com/drive/folders/19X68JeiXdeZgFp3cuCVpac4aLLw4StHZ?usp=sharing (accessed on 9 January 2022).

**Acknowledgments:** This work was supported by [the Network Collaborative Manufacturing Integration Technology and Digital Suite Research and Development Project of the Ministry of Science and Technology] grant number [2020YFB1712401]; [the Key Scientific Research Project of Colleges and Universities in Henan Province] grant number [21A520042]; [Major public welfare projects in Henan Province] grant number [201300210500]. We thank LetPub (www.letpub.com (accessed on 9 January 2022)) for its linguistic assistance during the preparation of this manuscript.

**Conflicts of Interest:** The authors declare no conflict of interest.

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
