# Peer review of "PTRNet: Global Feature and Local Feature Encoding for Point Cloud Registration"

_applsci, doi:10.3390/app12031741_

Round 1
Reviewer 1 Report
This paper proposed a new end-to-end PCD registration model and compared its performance with other models through data application.
please supplement the following and submit the revised version.
1. The performance analysis result(as a value) of the PTRNet model is required in the Abstract.
2. Please clearly describe the scope and method of this study in the Introduction.
3. It is necessary to describe in detail the ModelNet40 mentioned in [chapter 3]. Please describe why the authors used this model.
4. It is necessary to describe in detail what Fig. 3 means.
5. It is necessary to an explanation for Table 1~4. The reason for the difference in the result value of each model should be explained.
6. The limitation of this study should be described. It is not suitable to simply mention the excellence of the proposed model.
7. Please clearly describe the future research.
8. [Chapter 5] should summarize the results of this paper and includes limitations and future research directions. Please revise it.
Reviewer 2 Report
I believe that this paper is presenting some good work on the registration of point clouds for finding transformations. The paper presents a new method where a point transformer is used for combining global and local features with a KNN technique to improve performance. The paper compares this method with several state-of-the-art methods from the literature on the topic and shows that the new methods provides better results than the existing methods. I believe that the paper's work is solid and of interest to many readers. I only have a few suggestions as to how the paper could be improved:
- The field of models for which point registration could be applied is very large. So the readers could better determine if the proposed method could be applied to their specific problem, it would be nice to have more information on the types of models in the training and evaluation sets used in this paper. Were the models all man-made objects such as furniture? Were any living things tested? Were all surfaced hard or were objects with flexible surfaces such as animals used? Was the method checked with curved surfaces?
- Several proposed methods are compared with the newly proposed method. It would be nice to see which of these methods are closest to the newly proposed method. The paper claims that the use of a KNN topology with attention network is responsible for the improvement. Are there any methods which are similar to the proposed method without this KNN topology/attention network? That is, is there evidence that the improvement is coming from the proposed improvements?
- It would be nice to have some information on the training time and computational cost of the proposed method in comparison with the methods described in the previous papers.
In summary, the paper's main work is solid. With the additions of the information above, the paper should be ready for publication.
Round 2
Reviewer 1 Report
Please supplement the conclusion of Chapter 5.
Author Response
Point 1: Please supplement the conclusion of Chapter 5.
Response 1: The conclusion of Chapter 5 has been supplemented in the new edition of the manuscript.